# Exploring the Determinants of Food Choice in Chinese Mainlanders and Chinese Immigrants: A Systematic Review

**DOI:** 10.3390/nu14020346

**Published:** 2022-01-14

**Authors:** Yixi Wang-Chen, Nicole J. Kellow, Tammie S. T. Choi

**Affiliations:** Department of Nutrition, Dietetics and Food, School of Clinical Sciences, Monash University, Melbourne, VIC 3168, Australia; ywan0583@student.monash.edu (Y.W.-C.); nicole.kellow@monash.edu (N.J.K.)

**Keywords:** food choice, traditional Chinese medicine, Chinese immigrants, food behaviour, Chinese culture, harmony, environmental factors

## Abstract

Determinants of food choice in Chinese populations have not been systematically synthesised using a cultural lens. This study reviewed qualitative studies exploring food choice determinants of both Chinese mainlanders and Chinese immigrants living in Western countries. Ovid Medline, CINAHL Plus, Web of Science, ProQuest, and China National Knowledge Infrastructure database (CNKI) were searched from database inception to 1 April 2021. Studies were included if they involved qualitative research methods, were written in English or Chinese, investigated the factors influencing food choices, and targeted Chinese mainlanders or Chinese immigrants living in Western countries. Twenty-five studies (24 in English, 1 in Chinese) were included, involving 2048 participants. Four themes were identified; (1) the principles of traditional Chinese medicine (TCM), (2) perceptions of a healthy diet in Chinese culture (e.g., regular eating, eating in moderation, and emphasis on food freshness), (3) the desire to maintain harmony in families/communities, and (4) physical/social environmental factors all significantly influenced Chinese people’s food choices. It is important to acknowledge these factors when developing culturally appropriate nutrition programs for promoting health in Chinese mainlanders and Chinese immigrants.

## 1. Introduction

Non-communicable chronic diseases (NCDs) kill 41 million people worldwide each year, equivalent to 71% of all deaths globally [1]. Diet is a major contributing factor to the development of many NCDs [2]. The pathophysiology of common diet-related diseases, including overweight/obesity, type 2 diabetes, gastrointestinal disorders, cardiovascular disease, and certain cancers, is related to excessive consumption of ultra-processed foods high in saturated fat, salt, and sugar but low in dietary fibre, vitamins, minerals, and antioxidants [3]. In response to the exponential rise in NCDs, most countries have developed healthy dietary guidelines to educate and encourage people to adopt a balanced diet. Why the majority of the population do not adhere to these guidelines is a core research question in the public health nutrition field. In order to answer this question, researchers need to first identify and understand the intrinsic and extrinsic factors influencing people’s food choices.

Food choice refers to the decision regarding what food to eat, when to eat, where to eat, how much to eat, how to eat, and with whom [4]. It is clear that eating is about more than health [5]. Eating is interrelated to many of life’s social functions and is a source of pleasure [5], which is an important reason why people do not always follow a healthy dietary pattern. An individual’s food choice is motivated not only by personal preference, but also by external influences such as time constraints, affordability, availability, local and national food policies, social and cultural practices, and food advertising. Contento [5] synthesised determinants of food choice into a framework that categorises influences on food choice into 10 sections, including biologically determined behavioural predispositions, experience with food, physiological and social conditioning, intrapersonal and interpersonal factors, and social/physical/economic/informational factors.

The myriad of factors influencing food choice and diet-related behaviours are highly complex and impossible to replicate in an artificially controlled research setting. Therefore, qualitative research methods are the most appropriate for exploring food choice determinants, because they are used to investigate how and why people behave in certain ways in their natural environment [6].

The increasing incidence of diet-related diseases in China is alarming. The percentage of overweight/obese Chinese adults continues to increase, from 29.9% in 2002, 42.0 % in 2015 [7], to 50.7% in 2019 [8]. The prevalence of diabetes has also followed a steady upward trajectory, from 0.67% in 1980, 2.12% in 1994, 4.5% in 2002, 9.7% in 2008, 10.9% in 2013 [7], to 11.2% in 2017 [7]. There were 245 million Chinese people diagnosed with hypertension in 2018 [9]. Nearly nine in ten recorded deaths in China were caused by NCDs in 2019 [8]. Along with the rise in NCDs, the dietary pattern of Chinese people has undergone dramatic changes over the last three decades [10]. The main features of the dietary transition are the insufficient intake of fruits/dairy/whole grains and overconsumption of oil/fat/salt [11]. Lack of knowledge about healthy eating may be one contributor to the national increase in poor nutritional intake. However, rapid economic growth, urbanisation processes, and the wide availability of ultra-processed foods also contribute to the formation of nationwide unhealthy dietary patterns.

Chinese immigrants living in Western countries have an elevated risk of developing chronic diseases over the course of their acculturation [12,13], higher than that of mainland Chinese who remain in China [14]. The cause of this phenomenon has been frequently investigated. A longitudinal study revealed that acculturation increases with length of residence in the USA and is accompanied by an increase in the energy density of the diet [15]. Similar results were obtained from a cross-sectional study which found that high dominant societal immersion in America was positively associated with higher red and processed meat consumption, and lower intakes of vegetables, fruit, and legumes [16]. Studies involving Chinese Australians and Chinese Canadians have also shown similar phenomena [12,17]. Common unfavourable dietary changes post-migration have included a high intake of sugar-sweetened beverages, larger portion sizes, and increased consumption of highly processed convenience foods [16,17]. It is also true that Chinese immigrants adopt some healthy dietary changes in their host country, such as increased intake of whole grains [16]. The retention of traditional Chinese dietary patterns and the adoption of only the healthy features of the Western diet may help to reduce the risk of chronic disease in Chinese immigrants. In order to encourage Chinese immigrants to make a positive dietary transition during acculturation, researchers, health educators, and policymakers need to clearly understand the determinants of food choice in Chinese immigrants to facilitate the design of appropriate ethnic-specific nutrition interventions.

Food plays a central role in many aspects of Chinese society and is more than simply the components required to meet physiological needs. Food is used to establish and maintain interpersonal relationships, celebrate important events, and represent social status [18]. Certain foods also develop symbolic meanings, such as eating long noodles for longevity when celebrating birthdays and eating glutinous rice balls for cohesive united family members [18]. Traditional Chinese medicine (TCM) was previously the dominant theory instructing Chinese people to eat a yin/yang balanced diet. Despite Western nutrition science being widely accepted by Chinese society today, the hallmarks of TCM’s continuing influence on people’s diets are easy to identify, e.g., preferring warm water to cold water, adding herbal ingredients to make therapeutic cuisines [19], the value placed on certain foods (bird’s nest, shark’s fin, bear’s paw) that are not considered edible in other nations. The distinct Chinese dietary norms combined with the impacts of globalisation have shaped the current diet culture for Chinese people. It is important to explore “why” Chinese people eat what they do.

### Aim

No previous systematic reviews have focused on Chinese mainlanders and Chinese immigrants to explore their food choice determinants using a cultural lens. This review aims to synthesise the outcomes of published qualitative research exploring the determinants of food choice in Chinese adults living in mainland China and Chinese immigrants living in Western countries.

## 2. Method

### 2.1. Study Identification

This review was conducted in accordance with the preferred reporting items for systematic reviews and meta-analyses (PRISMA) [20]. The review was prospectively registered on a systematic literature review registration website (PROSPERO 2021 CRD42021250223). A systematic literature search was performed to retrieve publications on the determinants of food choice in Chinese mainlanders and Chinese immigrants living in Western countries. Two searches were conducted, an English-language literature search and a Chinese-language literature search. For English literature, four electronic databases (Ovid Medline, CINAHL Plus, Web of Science, ProQuest) were searched from inception to 1 April 2021. The English search terms used included Asian Continental Ancestry Group/OR Chin$ OR “Emigration and Immigration”/OR “Emigrants and Immigrants”/OR Asian Americans OR “Transients and migrants”/OR “Chinese immigra$”/OR “Chinese migra$” AND Food preferences/OR Feeding behaviour/OR “Food choice$” OR “Diet$ preference$” OR “Eating habits$”. The full search strategy is presented in Appendix A. For Chinese literature, we searched the China academic journals full-text database, which is also known as the China National Knowledge Infrastructure database (CNKI). The Chinese search terms were translated from the English keywords by two Mandarin-speaking members of the research team (YW, TC). Due to the differences in the language context between Chinese and English, some Chinese search terms were identified based on the keywords of relevant articles. For instance, the Chinese search keywords included “食物选择” + “饮食选择” + “膳食选择” + “食品选择” OR “饮食决策” + “膳食决策” OR “(饮食 + 膳食) * 喜好” OR “(饮食 + 膳食) * 偏好” OR “(食品 + 食物) * 购买意愿” (Appendix A). Both English and Chinese literature searches were limited to adult humans (18+ years), qualitative research (ethnography, interviews, focus groups, document analysis) involving Chinese participants (living either in mainland China or a Western country), with any health condition.

### 2.2. Screening and Eligibility

All resultant references were imported into a systematic review screening and data extraction software program, Covidence (Covidence Systematic Review Software, Veritas Health Innovation, Melbourne, Australia), which was used to screen studies and identify those meeting the pre-specified inclusion criteria. The Covidence program automatically identified and eliminated duplicate articles. During the first pass, article titles and abstracts were screened by YW and either NK or TC independently to determine their suitability for inclusion. Selected studies then underwent full-text screening, which was also conducted by YW and either NK or TC independently. Conflicts were resolved by discussion until consensus was reached. On completion of screening, the PRISMA Flowchart was automatically generated by the Covidence program.

### 2.3. Data Extraction and Quality Assessment

Data were extracted from each article using a data extraction table, which included: first author, affiliation, research methodology, theories or theoretical framework, sampling approach, participant characteristics, data collection methods, study aim, key themes, and author’s explanation of themes. Data extraction for all articles was completed by the first author (YW), with NK cross-checking the data extraction for ten randomly selected studies. For studies involving participants with multiple ethnicities, data were extracted which was specific to the Chinese participants only.

The Joanna Briggs Institute Critical Appraisal Checklist for Qualitative Research (JBI Checklist) [21] was employed to assess the research quality of retrieved studies. The JBI Checklist was designed to appraise the methodological quality and validity of a qualitative study and “to determine the extent to which a study has addressed the possibility of bias in its design, conduct, and analysis” [21]. The JBI Checklist focuses on congruity between the research question, methodology, data analysis and interpretation of results [22]. In this review, all the papers were assessed using the JBI Checklist in duplicate, by the first author (YW) and a second author (TC or NK) independently, with conflicts resolved by discussion until consensus was reached.

### 2.4. Data Synthesis and Analysis

Data from all retrieved studies were synthesised using a narrative approach, which relied mainly on the use of texts/words to synthesise and to interpret the findings of the included studies [23]. Guidance on the conduct of narrative synthesis in systematic reviews designed by the Economic and Social Research Council (ESRC) Methods Programme was followed [23]. Firstly, Contento’s framework of food choice determinants [5] was used as a guide to identify different categories of food choice determinants, of which we paid attention to culturally related determinants. Secondly, the first author, YW, developed a preliminary synthesis to organise the determinants into groups, then conceptualised initial patterns across the groups. Thirdly, the other two authors, TC and NK, cross-checked the robustness of the synthesised patterns until consensus was achieved.

## 3. Results

Initial database searches yielded a total of 5535 citations. Following the removal of duplicate articles that did not meet the inclusion criteria, 4862 citations remained (3744 in English, 1118 in Chinese), of which 25 studies (24 in English, 1 in Chinese) were included for qualitative synthesis (Figure 1). All 25 studies were published between 2000 and 2021.

The number of Chinese participants in each study ranged from 1 to 396. The studies were conducted in the USA (8 papers), China (8 papers), Canada (4 papers), Australia (3 papers), Spain (1 paper), and the UK (1 paper). Semi-structured interviews were the most common data collection method utilised (17 of 25), followed by focus group (9 of 25), ethnography/observation (5 of 25), and document analysis (2 of 25). The characteristics of all included studies are shown in Table 1.

Nine of the 25 studies used a combination of data collection methods, for example, a combination of observation, semi-structured interview, and focus group [30,42], or a combination of focus group and semi-structured interview [24,40], or a mix of observation and key informant interviews [26,39,45]. The remaining studies used a single qualitative data collection method.

There were 2048 participants involved in the included studies, and approximately 69% were females (calculated from the available data). Study participants ranged in age from 18–90 years. The length of time spent living in Western countries by Chinese immigrants ranged from six months to 22 years. Eight studies specifically recruited older participants (≥50 years), five recruited young participants (≤30 years), and the remainder included adult participants of a variety of ages. Three studies recruited people with particular disease conditions (cancer or high blood pressure) [26,39,47], with the remainder of the studies investigating healthy people.

### 3.1. Assessment of Methodological Quality

As evaluated by the JBI Checklist, all the studies met the criteria for demonstrating congruence between the research methodology used and the research question, data collection methods, data analysis, and the interpretation of results, (Table 2) with the exception of one Chinese paper that provided very limited details on the methodology used [33]. A consistent weakness of all the studies was the lack of discussion regarding the authors’ cultural and theoretical positioning in the respective study. A small number of studies briefly described the researchers’ background (mainly bilingual ability or expertise) [24,26,30,31,34,40], but none discussed the researchers’ potential influence on the study. Only five studies used theories or theoretical frameworks to design the research [32,35,42,44,47].

### 3.2. Qualitative Analysis of Study Findings

#### 3.2.1. The Concept of Traditional Chinese Medicine Influences Participants’ Food Choices

TCM was a common factor influencing Chinese immigrants’ food choices, whereas in studies involving Chinese mainlanders, no participants mentioned TCM. For the studies involving Chinese immigrants, eight of 17 frequently described achieving a balance of yin/cold and yang/hot, a well-known principle in TCM [25,28,29,39,40,41,42,44]. Participants believed that it was important to balance yin and yang foods in order to maintain health [25,28,29,39,40,41,42,44]. For example, pregnant women avoided eating yin/cold foods (e.g., crabs, bananas, cold drinks) because they were considered to cause harm to the baby, according to TCM philosophy [25,29]. A study exploring factors influencing food choices of Chinese university students living in the USA reported that participants tended to eat more “yin” foods in the summer, more “yang” food in winter [44]. Moreover, participants believed that some food (e.g., herbs, garlic, ginger, swallow saliva, white fungus) had either healing or damaging properties [40,49,50].

Although the concept of TCM obviously shaped Chinese immigrants’ food choices, not all participants trusted TCM completely, especially when they encountered conflicts between dietary advice from TCM and Western nutrition science [26,29,39].

#### 3.2.2. Individual’s Perception of a Healthy Diet in Chinese Culture Influences Food Choices

Study participants’ food choices were not only affected by the concept of TCM but also their cultural perception of healthy diets. Generally, participants held perspectives that diet was closely associated with health [26,29,30,31,37,40,41,42,43] and that Chinese food was healthier than Western food [28,29,32,41] (studies did not specify how “Western food” was defined). Some of the participants’ views about healthy eating were in alignment with the orthodox Western dietary recommendations, such as eating more fruit and vegetables, avoiding food high in fat/oil, and eating a variety of foods [28,39,42]. However, other participants’ definitions of healthy eating were underpinned by Chinese culture. For example, “freshness”, as a requirement of healthy eating, was emphasised in five studies [24,28,29,38,42]. the concept of “regular eating” was observed in three studies [29,38,42], followed by “eating in moderation” seen in two studies [36,42].

Additionally, participants’ dietary and health beliefs were shaped by health information obtained from a wide range of sources, including family members, friends, relatives, newspapers, magazines, TV, health practitioners, Internet, fellow patients, dietitians, library books, community talks, and family physicians. [27,29,32,39,42]

#### 3.2.3. The Desire to Maintain Harmony in Family/Community Influences Food Choices

In family settings, the consideration of other family members was a common factor influencing participants’ food choices. For instance, young adult women reported eating unwanted food or overeating to avoid conflict and to show respect and gratefulness to the cook [34]. Chinese immigrants who grew up in China still preferred a Chinese diet after moving overseas [25,28,29,32,35,36,40,41,42]. However, in immigrant families with children, parents would also give in to the children’s desire to eat and buy Western food [24,28]. Some families even developed rules to balance everyone’s preferences [35,47].

Pregnant Chinese Canadian research participants reported following family members’ dietary advice to avoid conflicts, even when they did not really trust the advice [29]. In an auto-ethnographic study, an elderly mother chose to change her diet for better health, in order to reduce the future burden on her son to care for her [31]. In other social settings, Chinese immigrants living in an American nursing home prioritised group needs over individual needs [45]. They accepted the food provided by the facilities, including the foods they disliked [45]. In the older Chinese people’s view, if the food in Western hospitals was not culturally appropriate, other family members were expected to bring food to the hospital on a daily basis, rather than making a complaint to the hospital [40].

#### 3.2.4. Physical and Social Environmental Factors Influence Food Choices

Nine studies explored food choice determinants via direct interaction with participants (interview and/or focus group) and identified nine factors influencing food choices, including time constraints, cost, food quality, availability, convenience, taste, living conditions, family support, and cooking facilities [25,27,29,32,33,39,42,44,47]. Two studies conducted in China identified that food safety concerns also affected participants’ food choices. For example, participants preferred to purchase food from overseas [37] or eat in their worksite canteen rather than from outside restaurants [30] because they felt this food was safer.

Two document analysis studies collecting people’s food review microblogs, found that people living in geographically close areas, rather than climate-close areas, had similar cuisine preferences [46]. They also identified that humidity influenced diet preferences more than temperature [46]. Season, eating ambience, and food appearance also affected participants’ food preferences [48].

## 4. Discussion

The findings of this systematic review demonstrate that Chinese people’s food choices were influenced by both unique Chinese cultural factors and general environmental factors. TCM as a traditional Chinese philosophy affected participants’ food choices both directly and indirectly. Directly, participants’ perceptions of TCM caused them to concentrate on achieving a balance of yin and yang when choosing food. Indirectly, many of the foundational concepts of TCM have evolved into the social norms of healthy eating in Chinese culture, such as regular eating, freshness, and eating in moderation. Chinese immigrants showed a more obvious consideration of TCM when making food choices, compared to Chinese mainlanders. The desire to maintain harmony in families or communities played an important role in food selection for study participants, represented by their descriptions of eating unwanted food or overconsuming food in families, or prioritising group needs over personal needs within communities. Physical and environmental factors, such as time constraints, cost, food quality, food safety, convenience, taste, living conditions, family support, climate, season, eating ambience, and food appearance were also found to have a significant influence on food choices.

### 4.1. The Concept of TCM Influenced Chinese People’s Food Choice

TCM has a unique cultural influence on Chinese people’s dietary behaviours. While individuals may have different levels of knowledge of TCM [49], many generally believe in the concept of achieving a nutritional “balance of yin and yang”, despite the categorisation of yin/cold and yang/hot foods being inconsistent in different publications [50]. For example, mango is labelled “cool” in the Encyclopaedia of Chinese Diet, while it is regarded as “plain” in the Compilation of Chinese Herbal Medicine [50].

A cross-sectional survey conducted in Shanghai in 2013 showed that almost 50% of older adults believed in and used TCM, with people of higher socioeconomic status (SES) tending to use TCM to complement conventional medicine, while those of lower SES used TCM as an alternative to conventional medical treatments due to its lower cost [51]. TCM remains popular because it has a focus on the care of the “whole person” and the provision of quality-of-life care when a cure is not possible. Given that TCM principles include the tenets “medicine and diet are homologous” and “food is not medicine but better than medicine” over thousands of years [52], it is understandable that TCM would influence food choices in Chinese people. Moreover, given the increasing prevalence of NCDs worldwide, it is speculated that Chinese people may integrate TCM into their diet to reduce the risk of NCDs or relieve symptoms. However, most published research focuses on investigating the popularity of TCM [53], the efficacy of TCM on NCDs [54], and general perceptions about TCM [55]. There is a lack of research exploring how TCMs, used to prevent and treat NCDs, influence food choice.

An interesting finding of this review was that Chinese immigrants living in Western countries mentioned TCM in their food choice decision-making more frequently than Chinese mainlanders. Five reasons are speculated to contribute to this difference. Firstly, this review included 17 qualitative studies recruiting immigrants living outside of China, whereas only 8 studies researched Chinese mainlanders. Two studies conducted in China used a document analysis approach which only examined participant food reviews and did not provide a complete picture of food choice determinants [46,48]. A third study (conducted in China and written in Mandarin) was determined to be of poor methodological quality due to its very limited information about participant recruitment, interview methods, and data analysis approach, and did not adequately report the participants’ determinants of food choice [33]. Therefore, the Western-dominant profile of included studies may have contributed to the disparate findings regarding TCM. Secondly, previous research has demonstrated that the Chinese elderly trust TCM more than younger Chinese [51,56]. In this review, the majority of studies (13/17) conducted in Western countries solely or partly recruited elderly participants. In contrast, only 3 of the 7 studies conducted in mainland China included the elderly. Larger numbers of elderly Chinese research participants in the studies conducted overseas would therefore be expected to report a greater focus on TCM. Thirdly, in contrast to Chinese mainlanders, the Chinese immigrants valued TCM not just as a resource for disease prevention and treatment, but also as a mechanism to reaffirm their cultural identity as Chinese [57]. For example, a study found that in the Chinese American community, TCM is part of the Chinese cultural heritage that extends beyond medical clinic visits [58]. This phenomenon is not unique to Chinese immigrants, with studies reporting that migrants from a variety of countries continue to adhere to traditional dietary practices as a means to preserve their cultural identity even after these practices have lost popularity in their countries of origin [59,60] Fourthly, as globalisation has increased in recent years, the extent to which members of the same ethnic group attach their lives to their traditional culture appears to be ever-changing [61]. Rapid economic development and the current global prominence of the Western medicine system may result in young Chinese people having fewer opportunities to be exposed to TCM, leaving only the elderly to trust and utilise TCM. Lastly, Western researchers studying Chinese immigrants may place more emphasis on culture as an influencer of food choice, whereas researchers investigating Chinese mainlanders are predominantly Chinese themselves, who were born and grew up in China, and may tend to overlook or downplay the importance of cultural influence. Further qualitative study should investigate the current influence of TCM on food choices in Chinese mainlanders.

TCM and conventional Western nutrition medicine are two entirely dissimilar theory systems [52]. The contrast in philosophies, fundamentals and terminologies between the two approaches often impede the integration of both fields of knowledge [52]. This inevitably causes confusion for Chinese people who trust in TCM when they come across contradictory dietary advice from both systems [26,29]. There is no authority resource the public can consult to resolve their confusion, and health care professionals are unlikely to provide integrated TCM and Western nutrition advice. Given the profound influence of TCM on food choice, it is necessary to make efforts to integrate TCM and Western nutrition science for the benefit of the Chinese population [62], as recommended by the World Health Organisation (WHO) in their Traditional Medicine Strategy 2014–2023 [63].

### 4.2. Perceptions of a Healthy Diet in Chinese Culture Are Considered When Making Food Choices

Three cultural perspectives about healthy eating, including “regular eating”, “freshness”, and “eating in moderation” were frequently reported in the included studies. The authors of these studies have appeared to simply translate participants’ Chinese words into English but did not explain these three concepts in detail. The contextual meanings of these three concepts in China are much more complicated. There are many old expressions related to them in Chinese folk culture, such as “若要身体安, 三分饥和寒” (keep yourself a little bit cold and your stomach a little bit hungry, for better health), “饮食有节” (eat at regular times and in regular amounts) [64] “吃饭七分饱, 健康活到老” (be 70% full after every meal, you will age healthily), ’应时饮食’ (eat seasonal fresh foods), ‘早餐吃的像皇帝, 午餐像平民, 晚餐像乞丐’ (eat breakfast like a king, lunch like a civilian, and dinner like a beggar). These folk cultural dietary concepts seem to have been inherited from TCM over the years, and subsequently incorporated into the beliefs and norms expected in Chinese society. For example, in the theory of TCM, freshness is of the highest importance, as fresh foods contain the most *qi* which can ideally develop their “specific thermal effect” [64]. At the very beginning of the classic TCM book, Huangdi Neijing (written over 2000 years ago), the authors point out that eating and drinking in moderation is one of the essential elements for longevity [65]. This book also explains that overeating damages the digestive system, and irregular eating leads to obesity/overweight [66]. Given the long history of TCM in China, it is understandable that “regular eating”, “freshness”, and “eating in moderation” have easily been adopted by Chinese people as rules to guide a healthy diet. These perceptions may be particularly important for Chinese people when making food choices.

Diet and health information is now obtained from a wide variety of sources, which may shape individuals’ perceptions of eating and influence food choices. In the studies included in this review, sources of dietary information included family members, friends, relatives, newspapers, magazines, TV, health practitioners, Internet, fellow patients, dietitians, library books, community talks, and family physicians [27,29,32,39,42]. People now employ a variety of social strategies to gather health information [67]. However, the information coming from multiple sources is often contradictory. In users of complementary and alternative medicine (CAM) for example, a survey found that CAM users reported better health outcomes from the treatments that are suggested by both professionals and friends/family/co-workers, compared to those treatments recommended by only professionals or only friends/family/co-workers [68]. Regarding healthy eating, the individual’s lay knowledge generated in society plays an important role in influencing their food choices [69]. Macintyre et al. for example, discovered that media reporting of food contamination scares had more influence on the consumption of eggs and beef in British people than the U.K. national guidelines for preventing coronary heart disease [69]. Therefore, health professionals should be aware of people’s lay knowledge and find appropriate ways to address it.

### 4.3. The Desire to Maintain Harmony in Families/Communities Plays a Role in Making Food Choices

Family harmony is a core element in Chinese society due to the influence of Confucianism over 2000 years. Confucianism encourages people to practice ren 仁 (benevolence, humanity) and Li 礼 (norms of proper social behaviour) [70]. People are obligated to live their lives in obedience to all the norms set by social institutions [70]. By following 仁 (Ren) and 礼 (Li), the society would be in a harmonious relationship within families and communities [70]. This long-lasting collective perspective has a profound influence on Chinese people. It also seems to affect Chinese people’s behaviour regarding their food choices. As demonstrated by the research included in this review, participants reported eating unwanted food or overeating to avoid conflicts and keep harmony in families/communities. This is consistent with a qualitative study investigating 20 Chinese patients (40–84 years old) with type 2 diabetes conducted in 2019 [71]. The study found that almost half of the participants ignored their physical needs to avoid troubles/conflicts in families, resulting in overeating food or eating the wrong foods [71]. Similarly, a study conducted in Taiwan involving 58 older adults revealed that 40% of participants chose to follow the family eating rules and self-sacrificed for the family [72]. These findings were exemplified by the participant quotes: “I don’t like the food my daughter-in-law prepares, but I eat it and never fight with her”, “I put my favourite foods aside for the needs of my family”, and “I try to eat and like the foods that my family likes” [72].

Seeking harmony within families does not necessarily impede healthy food choices. Instead, it can be a positive factor to promote healthier food choices. For example, some participants insisted on eating a healthy diet to keep fit, in order to avoid requiring care from other family members [31,73]. It seems that only when people conflate “harmony” with “the absence of conflicts”, that the perception of “harmony” becomes a barrier to healthy food choices. Conflicting ideas and preferences are not necessarily negative if people hold a positive view of them. Simply avoiding conflicts may reduce the chance of solving problems in a more appropriate way. An open and honest discussion within the family may or may not evoke tension, but at least it allows family members to acknowledge each other’s voices. In terms of food choices, individual’s overeating or eating unwanted food chronically instead of effectively solving the underlying problem is likely to lead to superficially harmonious family/community environments and poorer health outcomes.

The 2020 Nutrition and Chronic disease Survey (China) demonstrated that over 50% of Chinese adults are overweight/obese [8]. Death caused by chronic disease accounts for 88.5% of total deaths [8]. It should become the “new normal” for specific family members to make distinct food choices and eat in different ways, in order to achieve mutual health within a family. A new social norm in China should be encouraged which contends that harmony within a family is not threatened if someone does not eat the same food as others.

The studies in this review supported the concept that individuals are not isolated. Everyone is affected by others. In terms of food choices, family members are likely to be crucial influencers. In China, communal dining styles are dominant where people eat together and share dishes with others [18]. Choosing what to eat in a Chinese family is rarely a personal decision. It reminds healthy eating promoters that nutrition education may be more effective when designed for the whole family rather than just for individuals.

### 4.4. Physical and Social Environmental Factors Affected Chinese People’s Food Choices

This theme was similar to the social/environmental determinants of food choice in Contento’s framework [4] and included factors such as time, price, food quality, availability and convenience as major influences of food choice [4]. While these factors were similar in Chinese mainlanders and immigrants, an exception was “concerns about food safety” which had a more profound effect on Chinese mainlanders’ food choices. It has been well-documented that a series of high-profile food safety scandals in the past few years have seriously challenged Chinese people’s confidence in food safety. This may explain the findings of a meta-analysis that reported that Chinese consumers had the highest willingness to pay for food safety attributes in dairy product [74]. A 2016 study, exploring 142 Chinese mainlanders from 29 provinces of China on their attitudes about food grain safety found that public concern about grain safety has become a social phenomenon [75]. For example, this study revealed that 42% of respondents showed concern about the safety issues in grain production, and 73% were concerned about risks in the grain processing [75]. Similarly, another survey investigating 2092 consumers’ attitudes in six cities in China towards Fuji apple products revealed that about 43% of respondents were very concerned about food safety issues [76]. Food safety concerns have affected Chinese people’s food choice behaviours, with many choosing foreign grains [75] and purchasing certified products [76]. Therefore, food safety concerns have become a contemporarily new determinant of food choice for Chinese mainlanders.

Most of the qualitative studies in this review employed the in-depth interview as the main data collection method. It allows for the collection of detailed rich information. However, compared to the more naturalistic qualitative data collection methods (e.g., document analysis), using in-depth interviews has some drawbacks in exploring the determinants of food choice. Firstly, in-depth interviews identify participants’ conscious food choice determinants, such as affordability, time constraints, food quality, and availability. This is understandable because when participants are asked, they tend to figure out the logical reasoning behind their behaviour and do not consider what motivates their unconscious decision-making. Secondly, the limited interviewing time and the flowing nature of conversation may only allow the participants to articulate immediate or personal interest reasons/ideas, instead of encouraging deep, thorough thinking. Moreover, due to the nature of research, both researchers and respondents may be prone to be health-oriented when talking about food choices, which might introduce bias [77]. It is believed that cognitive thinking is a major factor that influences people’s food choices [78]. However, human behaviour also involves subconscious factors. Much of our intentional mental activity is not conscious at all [79]. In particular, making decisions about food choices on a daily basis can easily become a habit. Thus, subconscious factors also play an important role in shaping food choices [5]. The most common data collection tool used in qualitative studies, the interview, cannot explore subconscious factors involved in food choice decision-making. However, this review also included two studies that used document analysis as the data collection method [46,48]. By collecting Chinese people’s food review microblogs and analysing the text along with blogger’s demographic characteristics, the researchers discovered several influencers that were not identified by the interview-based method, such as, climate, temperature, and humidity. Food-choice determinants outside of participants’ immediate self-interest (e.g., food policy, country economic status, culture) were easily ignored or overlooked by participants during interviews. Hence, it is necessary to consider more than one data collection method to more thoroughly explore the complete picture of the determinants of food choice.

## 5. Strengths and Limitations

This is the first systematic review exploring Chinese mainlanders’ and Chinese immigrants’ food choice determinants identified by qualitative research approaches. Both English literature and Chinese scientific databases were searched and included papers written in both English and Chinese. However, there was only one eligible qualitative study identified that was published in Chinese. It is recommended that future researchers include English journal databases when investigating the determinants of food choice in Chinese people, as the number of relevant studies in Chinese databases is limited.

The studies included in this review employed a variety of data collection methods, such as in-depth interviews, focus groups, observation, and document analyses. These approaches complemented each other in this review, which allowed us to draw a fuller picture of the determinants of food choice.

The qualitative study critical appraisal assessment tool used in this review, the JBI Checklist [21], did not allow in-depth critique of the scientific rigour of included studies. For example, the Checklist does not contain criteria to assess important qualitative study features, such as data/research triangulation and data saturation. Instead, it focuses on the philosophy of the study and researcher and its congruity with methodology and methods [80]. Other available qualitative research assessment tools appear to have similar limitations. The JBI Checklist identified that most studies did not discuss the influence of the researcher on the research, and many did not include a statement of the researcher’s cultural and theoretical position. Failure to report the researcher’s positioning and reflexivity may have weakened the reliability of the findings in the included studies.

## 6. Conclusions

This systematic review synthesised published qualitative research investigating the determinants of food choice in Chinese people living in mainland China and Chinese immigrants in Western countries. This review found that the concept of TCM plays an indispensable role in Chinese people’s food choices, particularly in Chinese immigrants. Chinese society has developed unique cultural norms regarding the constitution of a healthy diet, such as “regular eating”, “freshness”, and “moderate eating”. These perceptions are often considered when making food choices. The desire to maintain harmony in families/communities had a predominantly negative impact on making food choices, leading to behaviours such as overeating and eating unwanted food. Food safety concerns, as a social environmental factor, were often considered by Chinese mainlanders. This review also indicates that using naturalistic data collection methods, such as document analysis can identify food choice determinants that are outside of participants’ conscious thinking. These findings advance the understanding of determinants of food choice in Chinese people and may be helpful for the design of culturally appropriate nutrition education and interventions.

## Figures and Tables

**Figure 1 nutrients-14-00346-f001:**
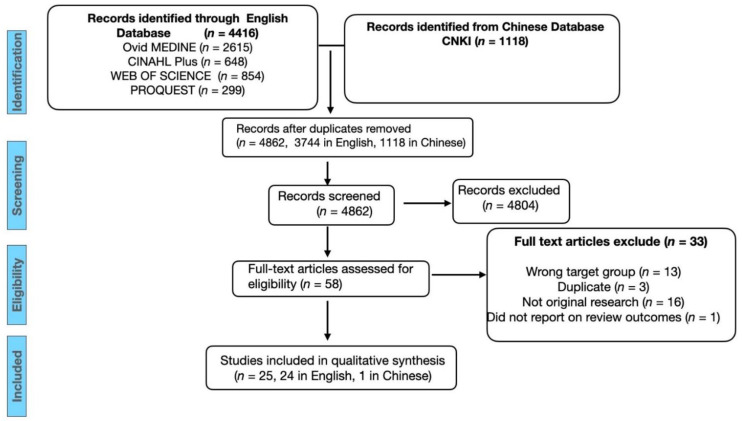
Flow diagram of the literature search. And screening results for a systematic review of the determinants of food choice in Chinese immigrants and Chinese Mainlanders.

**Table 1 nutrients-14-00346-t001:** Characteristics of included studies.

Study Title	First Author (Year of Publication)	Location	Sampling Approach	Data Collection Methods	Participant Characteristics	Summarised Findings
Changing Tastes: The Adoption of New Food Choices in Post-reform China [24]	Ann Veeck, A. (2003)	Nanjing, China	Purposive sampling	Structured observations; interviews with food shoppers and food retailers; focus groups with food shoppers. Field notes, photographs, information from the popular media in Nanjing.	Primary food shoppers (*n* = 20). Individuals (employees or entrepreneurs) who work in the food retail industry of Nanjing (*n* = 20). Three focus groups with people from different age groups: 25–35 y (*n* = 11), 34–35 y (*n* = 8), 46 y and older (*n* = 5).	There was an interplay of factors affecting Chinese consumers’ food choices. Increased income and time constraints lead to more processed food consumption. Participants also wanted to hold onto their traditional food consumption patterns, such as an emphasis on freshness, and cooking food to please family members.
Food Patterns Among Chinese Immigrants Living in the South of Spain [25]	Badanta, B. (2021)	Andalusia, Spain	Purposive sampling, snowball sampling	Face to face semi-structured interviews (15–30 min), field notes.	Chinese immigrants (*n* = 133), 61.7% F, 38.3% M, age range:18–55 y, average age: 30.7 y, living in Spain for average 11 y. Majority were ethnic Han, 78.3% from Zhejiang. 71% had medium-low educational levels.	Participants preferred a Chinese diet, but also integrated some Western foods. The concept of TCM and Chinese dietary norms influenced their food choices. The availability of Chinese food was not a problem in Spain, but long working hours limited their ability to eat healthily. Females and young participants were more concerned with healthy eating.
Perceptions of Food and Eating Among Chinese Patients with Cancer: Findings of an Ethnographic Study [26]	Bell, K. (2009)	City unspecified, Canada	Implied convenience sampling	Observations, key informant interviews.	Observations: Cantonese-speaking participants (*n* = 96), 36% M, 64% F, 61% cancer patients, 39% family members. Informant interviews: *n* = 7.	For the cancer patients, eating difficulties were often mentioned and were major concerns. They believed that the ability to eat well was very important for their health. The conflicts between Chinese and Western dietary recommendations confused the participants.
Built and Social Environmental Factors Influencing Healthy Behaviours in Older Chinese Immigrants to Australia: A Qualitative Study [27]	Cerlin, E. (2019)	Melbourne, Australia	Purposive convenience sampling	Nominal group technique sessions (*n* = 12), four sessions were designed for healthy diet	Participants (*n* = 91), 9 M, 18 F, age range: 60–85 y, average age: 71 y, living in Australia from 1–58 y (average 12 y); Most lived with their adult children.	There were 25 facilitators and 25 barriers for healthy diet identified. Facilitators: high food safety standards/regulations, receiving health education, family member support, educational information through Chinese newspapers or community talks, availability of healthy foods in grocery stores. Barriers: lack of family/household member’s support, financial restrictions, unhealthy food market environment. Wants: better provision of educational information, better access to grocery stores and fresh produce, improved public transport, fewer junk food outlets.
Seizing the moment: California’s opportunity to Prevent Nutrition-related Health Disparities in Low-income Asian American Population [28]	Harrison, G.G. (2005)	California, USA	Implied convenience sampling and purposive sampling	Focus groups (*n* = 24), and key informant interviews (*n* = 15)	Participants (*n* = 236):15 key informants, 116 adults (all parents), and 105 youths involved in focus groups. The adult participants were all low-income, first-generation immigrants.	Health beliefs (eating fresh fruit and vegetables, physical activity) were positive factors promoting healthier food choices. However, children’s adoption of American eating habits, the lure of fast food, and long work hours were barriers to following a healthy dietary pattern. Moreover, the trust in TCM concepts affected food choices.
Immigrant Women’s Food Choices in Pregnancy: Perspectives from Women of Chinese Origin in Canada [29]	Higginbottom, G.M.A. (2018)	Alberta, Canada	Purposive sampling	Women participated in one semi-structured interview, followed by a second photo-assisted, semi-structured interview which incorporated photographs taken by the women themselves.	Women (*n* = 23) who were in the perinatal period (from 5 months before to 1 month after childbirth), relocated from China to Canada within the past 10 years, age range: 26–42 y, average 31.6 y, highly educated, all had completed some form of post-secondary education.	The food and health related behaviours of immigrant females were influenced by their lay knowledge about health, cultural knowledge concerning antenatal and postnatal foods, TCM beliefs, social advice, health/nutrition information, and socioeconomic factors.
Enablers and Barriers to Improving Worksite Canteen Nutrition in Pudong, China: A Mixed Methods Formative Research Study [30]	Li, R. (2018)	Shanghai, China	Implied purposing sampling	In-depth interviews (*n* = 5), focus groups (*n* = 6), field observations	Community health centre administrators (*n* = 3) and canteen managers and staff (*n* = 3). Focus group participants: 19 M and 36 F employees, age 25–67 y. Male participants had a higher BMI than females.	Participants were proud of Chinese foods and cooks. They struggled with the balance between taste and nutrition. They felt that tasty foods were often unhealthy. Food safety concerns influenced participants eating behaviour (e.g., only eating in worksite canteen or bringing food to work).
Yang sheng, Care and Changing Family Relations in China: about a “Left-behind” Mother’s Diet [31]	Lin, X. (2020)	Mother in China, son (author) in the UK	Implied purposive sampling	Written communication between mother and son via WeChat	A Chinese migrant adult son (the author) living in the UK and his retired mother (in China)	Food practice intertwined with social and historical transformations. Yang sheng represented not only self-responsibility for health, but it also illustrated how intergenerational families show care and love for each other. (The author’s mother ate less to maintain her health, so that she would not be a burden on her son as she got older.)
Exploratory investigation of obesity risk and prevention in Chinese Americans [32]	Liou, D. (2007)	New York City, USA	Purposive sampling	In-depth interviews	Healthy US-born Chinese American adults (*n* = 40), 24 F, 16 M, age range 18–30 y, mean age 22 y, 60% full-time college students, 40% full-time employees.	Social environmental factors promoted unhealthy food choices (overeating) in Chinese Americans, including advertisements, inexpensive, and convenient fast food. They believed the traditional Chinese diet was healthier, but the degree of acculturation to the host country resulted in eating less Chinese food. Some traditional Chinese beliefs encouraged a slightly heavier physique.
影响大学生选择饮食消费场所的原因分析(Analysing the reasons of university students’ choice of eating location) [33]	Liu, W. (刘威) (2019)	Anhui, China	Implied convenience sampling	First, university students (*n* = 485) filled out the questionnaire. Second, researchers interviewed some students regarding the reasons for their daily food choices.	The number of students being interviewed was not identified. The students were undergraduates and studying either science or liberal art subjects at Anhui university. 40% F, 60% M.	Food safety and hygiene, time restrictions, cost, food taste, cuisine variety, and convenience of online shopping were factors influencing the undergraduate students’ choice of eating location.
The Clash of Culture and Cuisine: A qualitative Exploration of Cultural Tensions and Attitudes Toward Food and Body in Chinese Young Adult Women [34]	Liu, Y. (2020)	China online	Implied convenience sampling	Semi-structured interviews, via email (*n* = 27; 23 in English, 4 in Chinese) or Skype (*n* = 7, all in Chinese)	Adult women (*n* = 34), age range: 18–22 y, all had parents of Chinese descent, all were either currently living in mainland China or Hong Kong or had lived there for most of their lives (>10 y).	Participants encountered cultural eating norms (parents or grandparent pushing them to eat, obeying means a good child) and feminine appearance norms (being thin to achieve social acceptance). They developed different strategies to address these conflicts, either by fighting against the norms, accepting or ignoring the norms.
Chinese American Family Food Systems: Impact of Western Influences [35]	Lv, N. (2010)	Pennsylvania, USA	Convenience sampling	In-depth interviews (90 min). Couples were interviewed together first, then each partner individually.	Twenty couples (*n* = 40 individuals) with at least 1 child aged 5 y or older enrolled in a Chinese school in 1–3 sites in Pennsylvania. Age range: 21–65 y. Mostly dual-income, well-educated, middle-aged parents.	Chinese parents liked Chinese food, whereas children preferred Western food. When children got older, they started to appreciate Chinese foods. Fathers dominated the food choices of the family. Families had certain rules about food and used different strategies to balance everyone’s preferences, but often struggled to keep rules consistent. Many Chinese families’ diets were mainly Chinese style (Western breakfast, Chinese lunch and dinner).
Acculturation and Health behaviours among older Chinese immigrants in the United States: A Qualitative Descriptive Study [36]	Mao, W. (2019)	Los Angeles, USA	Purposive sampling	Face-to-face semi-structured interviews	Participant (*n* = 24), average age: 77 y, living in US for 22 y; 54.2% F, 45.8% M; participants were from mainland China, Hong Kong, and Taiwan. 70.8% spoke English, 54.2% spoke Cantonese at home. 54.2% reported good health	Older Chinese immigrants had limited interactions with other cultural groups. They depended on Chinese behaviour patterns and intra-ethnic networks. They exhibited strong maintenance of Chinese culture with some American cultural learning. Participants showed a strong identification with Chinese ethnicity contentment, fatalism, collectivism, individualism, and independence. Most followed traditional Chinese eating patterns or behaviours.
Innovations in the Agro-food System: Adoption of Certified Organic Food and Green Food by Chinese Consumers [37]	McCarthy, B. (2016)	Unspecified City, China	Convenience sampling	Structured questionnaire: open-ended survey questions designed to gather qualitative data	Respondents (*n* = 58), primarily F, age range: 35–44 y, living in tier 2 and 3 Chinese cities, married and with children. None were university educated, income 5000–8000 RMB/month.	Health and safety, rather than environmental considerations, were important determinants when making organic food purchasing decisions. Participants used a variety of strategies to deal with food related risks: buying food directly from farmers, purchasing overseas products, purchasing from formal channels. Information resources used: the Internet first, followed by friends and advertisements.
Exploring Meal and Snacking Behaviour of Older Adults in Australia and China [38]	Mena, B. (2020)	Frankston, Victoria, Australia	Implied convenience sampling	Three-stage focus groups: Stage 1: interview about meal and snacking behaviour, Stage 2: meat product tasting, Stage 3: perceptual mapping. Total length: 2 h.	Australians (*n* = 16) 13 F, 3 M, age range: 65–79 y; Chinese Australians (*n* = 21) 17 F, 4 M, age range: 60–81 y. All were meat eaters. They self-reported having good health, were physically active, and did not take any medication.	Australian participants did not eat breakfast and dinner regularly but snacked throughout the day. However, Chinese participants ate three meals regularly, and snacked only occasionally. Texture and flavour were key drivers of food choice for both groups.
Exploring the Dietary Choices of Chinese Women Living with Breast Cancer in Vancouver, Canada [39]	Ng, B. (2020)	Vancouver, Canada	Purposive sampling	Semi-structured interviews (60 min). Follow-up focus group with six participants	Chinese Canadian women (*n* = 19), age range: 41–73 y, diagnosed with breast cancer within the last 5 years. They were first or second-generation immigrants. Fifteen were married, most had children.	Breast cancer diagnosis resulted in dietary changes in participants: avoiding or restricting the consumption of certain foods, using TCM and natural health products.Obstacles to desired dietary change: the interplay between family life, personal food and social life, and work; the high cost and lack of availability of specialty foods, difficulties in assessing reliable and accurate nutrition information. Facilitators: support from family members.
Older Chinese People’s Views on Food: Implications for Supportive Cancer Care [40]	Payne, S.A. (2008)	Sheffield and Manchester, UK	Implied convenience sampling	Focus groups (*n* = 7), face to face semi-structured interview (40–50 min)	Stage 1: Participants (*n* = 46) were divided into 7 focus groups, Stage 2: Participants (*n* = 46) were involved in in-depth interviews. Older people were ≥ 50 y. Mean age: 66.25 y, 26 M, 66 F. Geographical origin: Mainland China (*n* = 49), Hong Kong (*n* = 41), majority self-reported poor health condition, 85% spoke Cantonese.	The older Chinese immigrants’ food beliefs: Food can be therapeutic, supportive, comforting, and prevent illness. Certain foods and cooking methods can be risky to health. These beliefs influenced their attitude toward the hospital food. They thought that foods in hospital were not culturally appropriate. Understanding the perceived cultural and therapeutic significance of food in hospital was important.
Intergenerational Transmission of Dietary Behaviours: A Qualitative Study of Anglo-Australian, Chinese-Australian and Italian-Australian Three-generation Families [41]	Rhodes, K. (2016)	City unspecified, Australia	Purposive sampling	Semi-structured interviews (40–60 min)	Three-generation families (*n* = 27), with ethnic backgrounds: Anglo-Australian (*n* = 11), Chinese-Australian (*n* = 8), Italian-Australian (*n* = 8). Average group interview size = 4, including at least one child (7–18 y), one parent, and one grandparent. Families were from middle class backgrounds. F 63.2%, M 36.8%	All families: Mothers and grandmothers dominated family food choice decisions, they influenced fruit and vegetable consumption by controlling purchasing decisions, insisting on consumption, reminding, and monitoring certain foods. In Chinese families: traditional culture played a large role in adult member’s food decisions. They believed the cultural diet was beneficial for health and wellbeing. The concept of TCM also influenced their food choice.
Use of Qualitative Methods to Study Diet, Acculturation, and Health in Chinese American Women [42]	Satia, J.A. (2000)	Seattle, USA	Implied purposive sampling	Observation of participants in their home kitchens, semi-structured in-person interviews (90 min), focus groups (2 focus groups, 2 h long each, 6 people each), 24-h dietary recalls	Interviews: 30 women, average age 51.9 y, 83.3% married, 67% had high school or lower education, 70% spoke little/no English, time spent living in the US: median 6 y. Focus groups: 12 women, average age 64.5 y, first generation Chinese Americans, 83% had low English proficiency, time living in the US: median 7 y.	Factors influencing food choice: Predisposing factors-traditional beliefs, taste preferences, beliefs about healthy eating, religion, existing dietary knowledge. Reinforcing factors-attitudes of friends, family members and health care providers. Enabling factors-convenience, cost, availability, quality/freshness.
Perceptions and Beliefs About the Role of Physical Activity and Nutrition on Brain Health in Older Adults [43]	Wilcox, S. (2009)	City unspecified, USA	Purposive snowball sampling	Focus groups (*n* = 42), of which, four focus groups included Chinese immigrant participants	Community-dwelling ethnically diverse older adults (*n* = 396), of which *n* = 36 were Chinese participants. 30.6% M, 69.4% F, age range: 50–90 y, 55.6% healthy weight, 38.9% overweight.	Chinese participants were more likely than Caucasians to be regularly physically active, eat fish at least once weekly, and to be within the healthy weight range. They believed that diet and physical activity help keep the brain healthy. Participants reported that portion control and healthy food preparation methods were important for brain health. They agreed that some types of foods should be eaten, and others avoided.
Acculturation and Environmental Factors Influencing Dietary Behaviours and Body Mass Index of Chinese Students in the United States [44]	Wu, B. (2016)	Chicago, USA	Implied convenience sampling	Focus groups (*n* = 7), each involving 4–7 participants, 24-h dietary recalls, food adoption scores, degree of acculturation, height and weight measures	Chinese students (*n* = 43) born and raised in mainland China, average length of stay in the USA: 20.3 months, 16 M, 27 F, age range: 19–31 y, 53% undergraduate, 47% graduate; Most lived in rented apartments. 19% M and 7% F were overweight/obese.	Extent of acculturation can predict Chinese students’ American food consumption. Having more American friends led to more exposure to American food. Living and cooking situation, and busy lifestyle were major factors influencing Chinese students’ food intake. Chinese students had difficulties in accepting raw, sweet, cold, or large portion sized American foods.
Hot Tea and Juk: The Institutional Meaning of Food for Chinese Elders in an American Nursing Home [45]	Wu, S. (2008)	City unspecified, USA	Implied purposive sampling	Meal observations, semi-structured interviews with residents (*n* = 7), family members (*n* = 9), and staff members (*n* = 17). Field notes.	Nursing home residents (*n* = 7), most were women, average age 81 y, all were immigrants, with the majority from Southern China, time living in the US: average 25 y. None spoke English fluently. Staff participants (*n* = 17): twelve were women, 5 men.	The American nursing home emphasised therapeutic personalised diets for the Chinese elders. They tried to provide Chinese food but lacked consideration of authentic Chinese ingredients and traditional presentation style, so failed to provide a meaningful Asian culturally appropriate diet. The Chinese elders prioritised community harmony over personal needs by either accepting the food provided or requiring family members to bring Chinese food to the nursing home.
Using social media to Explore Regional Cuisine Preferences in China [46]	Zhang, C. (2019)	China, online, Sina Weibo	Document analysis	Dish names (identified from Meishijie, a social media platform) were used as queries to retrieve related microblogs (food reviews) in Sina Weibo	There were 5156 cuisine names identified from 20 categories that represent Chinese regional cuisines. 3,209,990 cuisine reviews (personal microblogs) were retrieved.	Sichuan cuisine was most favoured among Sina Weibo users, followed by Shandong, Shanghai, Beijing, and Cantonese cuisine. The high-frequency dishes had lower regional differences among users. Geographical proximity was the key factor determining the similarity of regional dish preferences.
Facilitators and Barriers to Healthy Eating in Aged Chinese Canadians with Hypertension: A Qualitative Exploration [47]	Zhou, P. (2018)	Unspecified City, Canada	Implied convenience sampling	Telephone interviews (30–45 min), asking two open ended questions	Chinese Canadians (*n* = 30) with stage one hypertension; Mean age 60.8 y; living in Canada average length: 9.7 y, 16 F, 14 M, 66.7% participants lived a southern Chinese lifestyle.	There were facilitators and barriers at personal, family, community, and social levels that influenced healthy eating. Factors promoting healthy eating: experiencing positive effects of healthy diet, small family, supportive family, community health education workshops, printed educational materials. Factors impeding healthy eating: difficulty changing traditions, prioritising children’s wants, busy lifestyle.
Detecting Users’ Dietary Preferences and Their Evolutions via Chinese social media. [48]	Zhou, Q. (2018)	China online, Sina Weibo	Document analysis	25,675 dish names were used as queries to retrieve related microblogs. (Food reviews) in Sina Weibo	3,975,800 microblogs from 34 regions of China, reviews written by males (*n* = 1,207,909), females (*n* = 2,767,891), search date May 2015.	1. Popular dishes had fewer regional differences, while unpopular dishes had apparent regional restrictions. 2. Chinese users were most satisfied with taste, dish appearance, and service, the most unsatisfying aspect was function (effect on health). 3. Diners valued dining atmosphere more than taste. 4. Dining atmosphere and food appearance were important aspects for diners.

Abbreviations: *n*—number, F—Female, M—Male, y—years of age, TCM—Traditional Chinese Medicine, US—United States.

**Table 2 nutrients-14-00346-t002:** Quality assessment of included studies using the JBI Checklist.

First Author (Year) [ref.]	1. Is There Congruity between the Stated Philosophical Perspective and the Research Methodology?	2. Is There Congruity between the Research Methodology and the Research Question or Objectives?	3. Is There Congruity between the Research Methodology and the Methods Used to Collect Data?	4. Is There Congruity between the Research Methodology and the Representation and Analysis of Data?	5. Is There Congruity between the Research Methodology and the Interpretation of Results?	6. Is There a Statement Locating the Researcher Culturally or Theoretically?	7. Is the Influence of the Researcher on the Research, and Vice- Versa, Addressed?	8. Are Participants, and Their Voices, Adequately Represented?	9. Is the Research Ethical According to Current Criteria or, for Recent Studies, and Is There Evidence of Ethical Approval by an Appropriate Body?	10. Do the Conclusions Drawn in the Research Report Flow from the Analysis, or Interpretation, of the Data?
Ann Veeck, A. (2003) [24]	Y	Y	Y	Y	Y	N	N	Y	U	Y
Badanta, B. (2021) [25]	Y	Y	Y	Y	Y	N	N	Y	Y	Y
Bell, K. (2009) [26]	Y	Y	Y	Y	Y	Y	N	Y	Y	Y
Cerlin, E. (2019) [27]	Y	Y	Y	Y	Y	N	N	U	Y	Y
Harrison, G.G. (2005) [28]	Y	Y	Y	U	Y	N	N	N	U	Y
Higginbottom, G.M.A. (2018) [29]	Y	Y	Y	Y	Y	N	N	Y	U	Y
Li, R. (2018) [30]	Y	Y	Y	Y	Y	N	N	Y	Y	Y
Lin, X. (2020) [31]	Y	Y	Y	Y	Y	Y	U	N	U	Y
Liou, D. (2007) [32]	Y	Y	Y	Y	Y	N	N	Y	Y	Y
Liu, W. (2019) [33]	Y	N	Y	U	U	N	N	U	U	Y
Liu, Y. (2020) [34]	Y	Y	Y	Y	Y	Y	N	Y	Y	Y
Lv, N. (2010) [35]	Y	Y	Y	Y	Y	N	N	Y	Y	Y
Mao, W. (2019) [36]	Y	Y	Y	Y	Y	N	N	Y	Y	Y
McCarthy, B. (2016) [37]	Y	Y	Y	Y	Y	N	N	N	U	Y
Mena, B. (2020) [38]	Y	Y	Y	Y	Y	N	N	Y	Y	Y
Ng, B. (2020) [39]	Y	Y	Y	Y	Y	N	N	Y	Y	Y
Payne, S.A. (2008) [40]	Y	Y	Y	Y	Y	N	N	Y	Y	Y
Rhodes, K. (2016) [41]	Y	Y	Y	Y	Y	N	N	Y	Y	Y
Satia, J.A. (2000) [42]	Y	Y	Y	Y	Y	N	N	Y	Y	Y
Wilcox, S. (2009) [43]	Y	Y	Y	Y	Y	N	N	Y	Y	Y
Wu, B. (2016) [44]	Y	Y	Y	Y	Y	N	N	Y	U	Y
Wu, S. (2008) [45]	Y	Y	Y	Y	Y	N	N	Y	Y	Y
Zhang, C. (2019) [46]	Y	Y	Y	Y	Y	NA	NA	N	N	Y
Zhou, P. (2018) [47]	Y	Y	Y	Y	Y	N	N	Y	Y	Y
Zhou, Q. (2018) [48]	Y	Y	Y	Y	Y	N	N	U	U	Y

Abbreviations: Y—Yes, N—No, U—Unclear, NA—Not Applicable.

## Data Availability

Not applicable.

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
