# Peer review of "Exploring the Determinants of Food Choice in Chinese Mainlanders and Chinese Immigrants: A Systematic Review"

_nutrients, 2022, doi:10.3390/nu14020346_

Round 1

Reviewer 1 Report

The manuscript is well written. Nevertheless I would suggest to modify the tables of the results from vertical to horizontal specially table 1. 

It would be also interesting to describe the TCM in NCD that could be influence to food choice.

Author Response

  1. The manuscript is well written. Nevertheless, I would suggest to modify the tables of the results from vertical to horizontal specially table 1.

Author Response: We originally submitted the manuscript with Table 1 and Table 2 in horizontal (landscape) orientation, however the Nutrients editorial staff have changed them into a vertical (portrait) format.

  1. It would be also interesting to describe the TCM in NCD that could be influenced by food choice.

Author Response: Thank you for the advice. We have added some discussion regarding this point on page 18, Line 310-315.

Reviewer 2 Report

Title: suggest to use perception or influencing factors to replace determinants, as it did not use the keyword determinant in literature searching

the keyword used in English and Chinese were not exactly consistent, please explain the reason

move the keyword used to the methods part

Author Response

  1. Title: suggest using perception or influencing factors to replace determinants, as it did not use the keyword determinant in literature searching.

Author Response: Thank you for the suggestion. Although we did not specifically use the search term "determinants" in our search strategy, the search terms we used were developed to ensure the broadest scoping of literature to encapsulate the multiple factors influencing food choice. Therefore, we think the original title is appropriate.

  1. The keyword used in English and Chinese were not exactly consistent, please explain the reason.

Author Response: In order to minimise the automatic exclusion of relevant papers, the first 8 English search terms were not translated into Chinese for the CNKI search. The Chinese papers were then screened manually to include papers that reported the included populations of interest.

Regarding the search terms “Food preferences”, “Feeding behaviour”, and “Food choice” etc, initially, we translated those English search terms to Chinese directly, we then expanded them to ensure the broadest scoping of literature retrieved in CNKI.

  1. Move the keyword used to the methods part.

Author Response: Thank you for the suggestion, we have added the search keywords to the Method section. Please see Page 3, line 118-122, 129-131.